# A Depression-Risk Mental Pattern Identified by Hidden Markov Model in Undergraduates

**DOI:** 10.3390/ijerph192114411

**Published:** 2022-11-03

**Authors:** Xiaowei Jiang, Yanan Chen, Na Ao, Yang Xiao, Feng Du

**Affiliations:** 1Institute of Psychology and Behavior, Henan University, Kaifeng 475001, China; 2Department of Bioengineering, University of Pennsylvania, Philadelphia, PA 19104-6321, USA; 3Institute of Cognition, Brain and Health, Henan University, Kaifeng 475001, China; 4CAS Key Laboratory of Behavioral Science, Institute of Psychology, Chinese Academy of Sciences, Beijing 100101, China; 5Department of Psychology, University of Chinese Academy of Sciences, Beijing 100101, China

**Keywords:** depression-risk, Hidden Markov Model, coping style, emotion regulation, subjective well-being

## Abstract

Few studies have examined depression risk screening approaches. Universal depression screening in youth typically focuses on directly measuring the current distress and impairment by several kinds of depression rating scales. However, as many people have stigmatizing attitudes to individuals with depression, youths with depression were in fear of being known, and embarrassment held them back from reporting their depression symptoms. Thus, the present study aimed to identify the best, most easy access screening approach for indirectly predicting depression risks in undergraduates. Here, the depression score was ranked and viewed as the different stages in the development of depression; then, we used a Hidden Markov Model (HMM) approach to identify depression risks. Participants included 1247 undergraduates (female = 720, mean age = 19.86 years (std =1.31), from 17 to 25) who independently completed inventories for depressive symptoms, emotion regulation, subjective well-being (life satisfaction, negative and positive affect), and coping styles (positive and negative). Our findings indicated that the risk pattern (state 1) and the health pattern (state 2) showed distinct different rating results in emotional regulation, subjective well-being, and coping style. Screening for prospective risk of depression can be better accomplished by HMM incorporating subjective well-being, emotion regulation, and coping style. This study discussed the implications for future research and evidence-based decision-making for depression screening initiatives.

## 1. Introduction

In recent decades, researchers have pointed out a high prevalence of depression among undergraduates, and the number of youth suffering from depression has increased sharply, posing a grave threat to society [1]. On 10 October 2021, World Mental Health Day, the People’s Daily published a set of data. The detected rate of depression among teenagers in China was 24.6%, which showed a trend with increasing age, and some of the youth were chronically depressed. Depression was reportedly one of the most common mental disorders among college students in China, with an incidence rate between 15% and 35% [2]. Several explanations have been posited: undergraduates were confronted with numerous stressful life events inherent to adolescence and emerging adulthood, such as establishing their identity, emotional turmoil, achieving independence, entering unfamiliar environments, striving for academic performance, and selecting an occupation. As a result, undergraduate students around the world were at higher risk to develop mental health problems, particularly depression.

### 1.1. Factors Associated with Depression

A series of studies have revealed that an increased risk of depression in university students is correlated with several identifiable factors, such as age, gender, family socioeconomic status, living alone, family relationships, and academic worries with post-graduation life [3]. In a systematic meta-analysis investigated the risk factors of depression in medical students, the determinants of depression included individual factors, social and economic factors, and environmental factors [4]. In the Chinese context, existing literature showed a dominance of studies on depression among medical students and a scarcity of research among general college students. Furthermore, so far most work focused on sociodemographic risk factors for depression, with very few exploring other factors such as emotion regulation [5], coping style [6], and subjective well-being.

It has been proposed that depression was linked to difficulties in using emotion regulation strategies that actively serve to increase or maintain positive emotion. The susceptibility to depression was related with more frequent use of emotion suppression and less frequent use of reappraisal, which leads to the maintenance of negative emotions triggered by negative life events or cognitions [7]. Coping is a process in which individuals apply strategies (positive or negative) to help change the pressure circumstances or reduce the psychological stress associated with challenging situations, and one’s coping style to difficult experiences has significant implications for mental health [8]. A meta-analysis of studies on the association of coping styles with mental health found that a negative association between negative coping and psychological health. Specifically, a negative coping style has regularly been shown to be associated with increased depression among adolescents and young adults. In addition, negative coping appears to interfere with spontaneous improvement in minor depression [9]. Subjective well-being is a general term for people’s different assessments of life, events, and living circumstances, which relates to feeling good about one’s life, and included life satisfaction, positive affect (PA), and negative affect (NA) [10]. University students showed a significantly negative relationship between depressive levels and physical, social, functional, and affective well-being, and the measured levels of well-being were found to be significantly lower for depression participants [11].

In conclusion, many factors might increase college students’ susceptibility to depression. In addition, the prevalence of depression is very high, and the subsequent impact on this population is worrying, which may place them at risk for a variety of later more serious difficulties, including academic failure and dropout. The depressive symptoms of college students also have a profound impact on future careers, such as more absenteeism, reduced job performance, and even unemployment. It is urgent to develop strategies for early screening and management of depression in universities.

### 1.2. The Emergency of Depression Risk Screening

In the past decades, depression was a severe public health problem in youth and is associated with maladaptive outcomes, including impaired peer relationships [12], poor academic performance [13], and suicidal behaviors [14]. The economic and personal burden of depression was overwhelming, contributing to pervasive and chronic disability in occupational and interpersonal functioning [15,16]. The underdiagnosis of depression was a long-standing problem. A recent survey in a large urban area found that nearly half (45%) of major depression cases are not diagnosed [17]. Inadequate diagnosis of this disease has a considerable cost to quality of life and health care. In the quest to reduce this burden, measurable description characteristics of the risk of depression in individuals may prove helpful in targeting high-risk groups for prevention; it would also assist interventionists in better predicting depression risk and facilitating diagnosis to then focus on those at greater risk.

Increasingly, researchers and educators were encouraged to explore computational methods for the early screening and diagnosis of depression among university students, but detailed recommendations for how to accomplish this aim were largely missing. The typical way to offer mental health screening services on campuses was through appointments with counseling or health centers. Undergraduates seeking school counseling for their depression face significant limitations. Currently, many people in society retained negative attitudes toward people with depression and maintain increased social distance from them [18]. Moreover, students with depression can subjectively experience self-abasement, shame, and social withdrawal caused by stigmatizing attitudes. Their fear of others knowing about their depression and their embarrassment hold them back from asking for counseling help. In many places around the world, the public’s mental health literacy was still poor. However, objectively, it is not surprising that counseling center staff frequently report a strain on their available resources, with an average staff-to-student ratio of 1:2081, and larger campuses tend to report an even more significant burden, which contributes to longer waiting lists and shorter session limits [19]. Furthermore, many undergraduates experiencing minor depression symptoms that do not meet the diagnostic criteria for clinical depression would not receive medical treatment at a hospital [20,21]. Even though some depressive students asked for school counseling from their university, the university has no appropriate services for students suffering from depression; furthermore, it often takes a long time to see results.

### 1.3. The Present Study

Previous studies on individuals’ depression screening focused on the use of direct self-reported measurement to identify individuals with current depression. So far, there is no study to determine the population at risk of depression by indirectly evaluating related factors of depression. Low face validity assessments could better identify the depression risk since at-risk students may initially hide their depression symptoms during the consultation to simplify the undergraduate student counseling process with a lower psychological load. Consequently, measuring depression risk by another indirect questionnaire could be considered. The current study employed one such statistical framework, hidden Markov modeling (HMM) [22], to uncover depressive symptom states from self-reports of subjective well-being, coping style, and emotion regulation strategies to estimate participants’ probabilities of depression risk and to evaluate whether self-report variables predict participants’ depression risk state. HMM was one of the dynamic Bayesian networks [23] with the simplest structure [24], it is also included by the probabilistic graphical model (PGM) [25,26], and it can unsupervised cluster into k different states within every observation [27]. Findings from our study could provide an empirical foundation for feasible, multiformat depression screening initiatives and evaluate the relationship between mental states, such as emotional regulation, coping style, subjective well-being, and depression risk.

Generally, the stigma of depression and the overload of psychotherapy needs inhibit research from informing university college students’ depression risk screening when its use is encouraged. In addition, existing literature regarding the depression of college students mainly focused on medical students and a scarcity of research among general college students. Thus, there is a dearth of detailed analysis to associated factors of the depression states of college students, especially in a Chinese context. The primary aim of the present study is to bridge this gap in the literature by integrating current trends in youth assessment research to provide recommendations for college students’ depression screening. As well, we expect to use hidden Markov modeling to distinguish the different depressive symptom states according to the measured scores of subjective well-being, coping style, and emotion regulation strategies.

## 2. Materials and Methods

### 2.1. Participant

A total of 1247 university students without a present or prior history of medical or psychiatric disorders were recruited into the current study. A total of 120 participants were rejected as mistakes (>4 of 10) in polygraph items. A total of 1128 participants were eventually involved (age = 19.86 years (std = 1.31), from 17 to 25). The detailed Socio-Demographic characteristics of the sample are illustrated in Table 1.

### 2.2. Measurement

#### 2.2.1. Center for Epidemiological Studies Depression Scale (CES-D)

The Center for Epidemiological Studies Depression Scale (CES) [28] was first developed by Radloff in 1977 [29], Jie made its adaptation in Chinese in 2010 [28], and was used to measure the risk of depression. The scale is a Likert-type scale, and consisted of 20 items (e.g., “I’m worried about some trifles”). Each item used a 4-point scale ranging from 0 to 3, where 0 = none and 3 = always. The total score of the CES-D ranged between 0 and 60, which was summed by all items, and higher scores indicated greater levels of depressive risks. Generally, the CES scale which is higher than 20, was defined as the high depression group, while the health group scale was lower than 10. The Cronbach α of this Chinese version was 0.938 in the current study. By the confirmatory factor analysis (CFA), these indexes: χ2 = 1324.397, degrees of freedom (df) = 164, comparative fit index (CFI) = 0.902, root mean square error of approximation (RMSEA) = 0.079 were good.

#### 2.2.2. Subjective Well-Being Questionnaire (SWB)

The subjective well-being questionnaire (SWB) consisted of the positive affect and negative affect scale (PANAS) [30] and the life satisfaction scale [31]. As a 5-point Likert scale, PANAS had good reliability and validity in Chinese college students [32]. It included 20-item in two dimensions: positive affect (PA: 10 items, which were items 1, 3, 5, 9, 10, 12, 14, 16, 17, and 19, e.g.,” Interested”) and negative affect (NA: 10 items, which were items 2, 4, 5, 9, 10, 12, 14, 16, 17, and 19, e.g., “fidgety”) to describe the attitude toward life. After summing all items in one subscale, a Z-score was calculated for this subscale. The two subscales’ Cronbach α in the current study were 0.902 and 0.890, respectively. The Chinese version life satisfaction scale was developed by Xiong and Xu [33] as a 7-point Likert scale, including five items (e.g.,” My life is basically close to my ideal”). The Cronbach α in the current sample was 0.836. Further, the CFA results in this study (PANAS: χ2 = 1960.286, df = 169, CFI = 0.853, RMSEA = 0.097; Life Satisfaction: χ2 = 1960.286, df = 169, CFI = 0.853, RMSEA = 0.097) were accepted.

#### 2.2.3. Ways of Coping Questionnaire (WCQ)

The coping style questionnaire was based on the Ways of Coping Questionnaire (WCQ) by Folkman and Lazarus [34] and then simplified and modified by Jie [35] to be suitable for Chinese participants. The simplified WCQ included 20 items of two dimensions: positive coping (PC: composed items 1–12, e.g.,” Escape through work, study or some other activities”,) and negative coping (NC: composed items 13–20, e.g.,” Try to take a rest or vacation, and put aside problems (troubles) temporarily”). Each item was rated on a 4-point Likert scale (0 ‘never’ to 3 ‘very often’). The average score was used as the eventual score of each subscale. The retest correlation coefficient of the scale was 0.89. In the current study, Cronbach α was 0.90, while the positive and negative coping dimensions were 0.711 and 0.848, respectively. The CFA’s results of WCQ were χ2 = 1498.706, df = 169, CFI = 0.770, and RMSEA = 0.084.

#### 2.2.4. Emotion Regulation Questionnaire (ERQ)

The Chinese version of the Emotion Regulation Questionnaire (ERQ) [36] was used to collect emotion regulation strategies in this study. The ERQ was revised by Wang Li in 2007 [37] and consists of 10 items. It included two subscales: a cognitive reappraisal (6 items: 1, 3, 5, 7, 8, and 10, (e.g., “When I am in a bad mood, I will think of some happy things”)) and expression suppression (4 items: 2, 4, 6, and 9, (e.g., “Others’ criticism will make me sad for a long time”)). The two subscales are scored by adding up the scores of the composite items. In this study, Cronbach α is 0.710. The CFA results of ERQ were χ2 = 142.807, df =34, CFI = 0.968, RMSEA = 0.053.

### 2.3. Data Analysis

#### 2.3.1. Correlation

Two groups, the high-depression-risk group (CES score > 20, n = 411) and the low-depression-risk group (CES score < 10, n = 243) were classified by CES score. The Pearson correlation coefficient could show the strength and direction of the linear relationship between two variables, and the significance testing of the correlation coefficient provided whether it is significantly different from zero.

#### 2.3.2. Regression

Firstly, the depression-risk indicator, CES, and the other variables, positive coping, negative coping, positive affect, negative affect, life satisfaction, cognitive reappraisal, and expression suppression, were set in a linear regression based on the least square method in SPSS25.0 as dependent variable (y) and independent variables ($x_{i}$), respectively. The multicollinearity problem [38,39] of independent variables was detected by a variance inflation factor (VIF) [39] and tolerance:$$\begin{array}{c} Tolerance=\frac{1}{VIF}\; \end{array}$$

The value of VIF was greater than 1. Moreover, the closer the VIF value is to 1, the lighter the multicollinearity is, and vice versa. If the value of VIF < 10, there may be little multicollinearity [39]. The tolerance value was between 0 and 1. The small tolerance value means this independent variable was collinear with other independent variables, and there would be a significant error in the calculated value of the regression coefficient.

#### 2.3.3. Pattern Recognition by Hidden Markov Model (HMM)

The variables in the HMM consisted of two categories [24]. One was state variables$\;\left\{y_{1},\;y_{2},\;\ldots ,\;y_{n}\right\}$, where $y_{i}\in Y$ refers to the i’th state. It is usually assumed to be unobservable and hidden, so state variables are also called/named hidden variables. The other category was observed variables $\left\{x_{1},\;x_{2},\;\ldots ,\;x_{n}\right\}$, where $x_{i}\in X$ refers to the i’th observed variable. In the HMM, $y_{i}$ might have N possible values of states $\left\{s_{1},\;s_{2},\;\ldots ,\;s_{n}\right\}$, named the Gramma matrix. There are three critical assumptions for the HMM [40]: (1) the Markov Hypothesis (the state depends solely upon the previous state); (2) the Hypothesis (the state is independent of specific time); and (3) the Output independence hypothesis (the output is only relevant to the current state). The state is not directly visible, but some variables affected by the state are directly visible. Each state has a probability distribution over $x_{i}$. Thus, $x_{i}$ can reveal the independent state. The algorithm associated with the HMM is divided into three categories, each of which solves three kinds of problems: (1) Evaluation; (2) Learning; (3) Decoding. In this study, the HMM focuses on solving the decoding problem.

The Decoding process of the HMM is like rolling dice. Different personal characteristics ($x_{i}$) can be observed directly. However, the dice number (states, $s_{i}$) is not directly visible. Each rolling based on the various persons has a probability distribution of states ($y_{i}$), as shown in Figure 1. The HMM can go through different person characters to obtain the probability distribution of states, thus obtaining dice numbers.

Because of the preceding explanation, this study borrows the dynamic brain network analysis method in fMRI [41,42,43,44,45]. The order of participants was sorted by the CES, thus using HMM to recognize the depression risk pattern. We assumed that:$$x_{i}|s_{i}=k\sim Multivariate\;Gaussian\left(\mu_{k},\theta_{k}\right)$$
where k is the state number. $\mu_{k}$ is a vector with averaged observed variable elements, and $\theta_{k}$ is the covariance matrix of observed variables. Positive coping, negative coping, positive affect, negative affect, life satisfaction, cognitive reappraisal, and expression suppression were set as the observed variables ($x_{i}$). The number of hypotheses states (*k*) was 3. Estimating the HMM and the observed variable covariance matrix and the standard value of each state can be calculated by the HMM-MAR [41,42,43] toolbox in MATLAB R2020a.

## 3. Results

### 3.1. Descriptive Statistics and Correlation

Correlation results (Table 2) show that only positive affect and negative affect in these observed variables have low-level (the absolute values < 0.3) significant negative correlations with CES among all participants. In the low-depression-risk group, negative affect and cognitive reappraisal have low-level negative and positive significant correlations, respectively. However, no significant correlation with CES was shown in the high-depression-risk group. Among these observed variables, high-depression-risk and low-depression-risk, the relationships were complex at low-level and medium-level (0.3 < the absolute values < 0.6).

### 3.2. Regression

Regression results (Table 3) showed that positive affect (β = −0.22 ***), negative affect (β = −0.17 ***), and life satisfaction (β = 0.07 *) have significant regression relationships with CES among all participants (*F* (7,1119) = 16.282 ***, R^2^ = 0.087), However, no significant regression model was fitted in the low-depression-risk group (*F* (7,235) = 2.252, R^2^ = 0.035) and high-depression-risk group (*F* (7,403) = 1.825 ***, R^2^ = 0.014).

### 3.3. Pattern Recognition by HMM

These three states could effectively identify the depression risk group. Figure 2A shows the Z−scores of averaged observed variables among these three states. State 1 could be defined as a high-risk state, with low positive and negative effects. State 2, as the low-risk state, has higher positive affect, positive coping, and life satisfaction but lower negative affect, negative coping, cognitive reappraisal, and expression suppression than state 3, the medium-risk state. As given in Figure 2B−D, the covered range without the intersection of states 1 and 2 are [0,7] (nonrisk group: n = 188) and [23,60] (risk group: n = 319), respectively. State 2 has no exclusive covered range. Figure 2F showed the relationships of these observed variables in these three states by comparing among these three states.

The *t*-test is used to compare these two groups. The nonrisk group’s positive affect and negative affect were higher than the risk group, as shown in Table 4.

### 3.4. Comparison with Other Unsupervised Clustering Models

As shown in Figure 3, no clustering model can effectively detect the depression risk. Their clustering results do not provide any effective division in the CES score, like HMM in Figure 2C.

## 4. Discussion

To our knowledge, this study is the first investigation assessing the emotion regulation (cognitive reappraisal, expression suppression), subjective well-being (life satisfaction, negative and positive affect), coping styles (positive and negative), and the depression symptoms among undergraduate students in China. Therefore, it provides valuable insights into undergraduate students’ depression condition. Our results indicate that the assessment of emotion regulation, subjective well-being, and coping styles can indirectly reflect depression symptoms among undergraduate students.

In the current study, we first used HMM to identify depression risk by conducting indirectly related assessments and we contributed to the psychological characteristic description of depression. Similar to the range of the original CES from 10 to 20, we found thresholds of 8 and 22. Moreover, we assume that: the same or similar CES score has the hidden-related probability distribution of the states ($y_{i}$), as in the HMM hypothesis (1); every observed participant ($x_{i}$) is independent, as in HMM hypothesis (2); and every participant is only relevant to the current state ($s_{i}$), as HMM hypothesis (3). Based on these hypotheses, the questionnaire data as observed variables could be clustered by HMM in the current study. The risk pattern (state 1) and the health pattern (state 2) show different rating results in emotional regulation, subjective well-being, and coping style. Among them, the risk groups have both the lowest-level positive and negative emotions.

We found that college students with high–risk depression reported lower scores of positive affect and higher scores of negative affect than the non-risk group. This finding is consistent with several literatures on adults with depression that also show a significantly negative relationship between depressive levels and physical, social, functional, and affective well-being, and the measured levels of well-being were found to be significantly lower for depression participants [11]. In pilot studies, depression could also have a relationship with coping style [8] and emotion regulation [5], especially in younger people [6]. Many pieces of evidence suggest that depression is connected to hiding negative emotions by inert emotion fluctuation but is more reactive to positive events [46], thus causing sustained negative affect and loss of pleasure [47]. However, we were somewhat surprised to not find evidence for significant impairments in emotion regulation and coping styles as well. As described in the Introduction, the literature on adults with major depressive disordered suggests emotional regulation has been considered an important factor of depression; it is impaired in major depressive disordered patients [5], and the ability to recover from the ensuing negative effect is a significant difference between people with transient or severe depression. Likewise, negative cognitive processes and constructs are accessible in depression [48]. We speculated that there were probably two reasons for these differences. One possible reason is the different study participants and cutoff points used for depression symptoms severity categorization between different studies. Most previous evidence demonstrated difficulties in regulating emotions and frequent use of negative coping styles in major depressive disordered patients or medical students, while the present study focused on general college students. It is also possible that students who had major depressive disordered are no longer in the university, hence the number of those with severe depression is less in the present study. The other possible reason is the differences between eastern culture and western culture. Most Chinese universities have strict requirements for students’ entrance scores, but it is relatively easy after entering the university. And in the United States, college students are under tremendous pressure to meet graduation requirements, not entrance barriers. Therefore, American college students need to cope with higher academic pressure than Chinese college students.

However, the purpose of the present study is not to compare the difference in subjective well-being, emotion regulation, and coping styles between the high-risk depression group and the non-risk group. The contribution of this paper is that we used HMM to identify depression risk by conducting indirectly related assessments. Here, we focused on three factors associated with depression, which are subjective wellbeing, emotion regulation, and coping styles. Previous studies have shown that vulnerability to depression may be associated with frequent use of emotional suppression and infrequent use of reappraisal, leading to the maintenance of negative emotions triggered by negative life events or cognition. Increased emotional suppression is also linked to elevated levels of depression-related anhedonia [7]. It has been suggested that positive coping is associated with a lower frequency of depression, and negative coping has regularly been shown to be associated with increased depression risk among adolescents [9]. Another study showed that depression would be inversely related to subjective well-being [11]. Hence, these indirect indicators in our study are feasible to identify depression risk.

### 4.1. Implications

It is not just depression that plagues youths, it is depression stigma. The present study provided new thinking and indirect methods to address this issue. The higher the stigma of depression among college students is, the less likely they are to seek mental health care [49]. Fear of being labeled “crazy” by others is a major barrier to spontaneously reporting their depression symptoms and seeking mental health treatment [50]. Additionally, many students expressed stigmatized views, such as that experiences of depression may negatively affect their future job prospects and mate choices [51]. As an unsurprising result, many students with depression risk believed that it was useless to tell their parents. The results of our study showed a more stealthy and potentially even more effective approach to measuring depression risk in university students. According to the HMM approach, the students only need to answer some questions about their strategies of emotional regulation, life coping style, and subjective well-being; we can identify the depression risks of these students.

### 4.2. Limitations

Furthermore, the present study has limitations. First, the sample consisted of only university students, so the findings should be generalized with caution to middle school students and high school students. Second, some boundary CES scores, such as 8 and 22, are covered within the intersection of multiple states. Noise and an insufficient population might cause these intersections. Among the CES scores as an element in [8,22], the medium-risk group cannot be classified from the other groups, especially the high-risk group. These participants have unclear boundaries and complex mental performance. Third, due to the research time and funding constraints, several other important depression risk-related factors were not assessed, such as interpersonal relationships, early life stress, and parental depression. The results might have been different if more factors were included in the HMM. Finally, convenience sampling was used in recruiting subjects in the present study. This may create a sample bias, as students who volunteered for the survey may represent those with higher health awareness. Therefore, our results must be interpreted in light of these limitations.

## 5. Conclusions

The current study used HMM to identify depression risk by conducting indirectly related assessments of subjective well-being, emotion regulation, and coping styles and we contributed to the psychological characteristic description of depression. The present findings have substantial research and clinical implications. This study is the first to provide comprehensive screening information on indirect risk factors that can identify youths at high risk of depression; these results can inform the design and approach of targeted prevention efforts in the future. Indirect depression screening should be included in college students’ physical examinations, students’ mental health records should be established, and students with evaluation results showing risk should be given more attention. Moreover, further studies should consider these populations and focus on assessing and identifying healthy persons with some depressed emotions and unhealthy persons with depression risk. Also, in future research, more depression-related factors should be included.

There is a scarcity of studies on the screening of depression and its predictors in Chinese university students; moreover, associated factors were reported in major depressive disordered patients or medical students, and limited works were addressed in general college students in the previous studies. Therefore, screening the individuals susceptible to depression in college students is vital to investigate the problem thoroughly. Moreover, the results of the present study will provide a basis for university administrators and other stakeholders to screen and provide campus mental health services, and further help researchers interested in this field to conduct additional screening research on different factors associated with depression.

## Figures and Tables

**Figure 1 ijerph-19-14411-f001:**
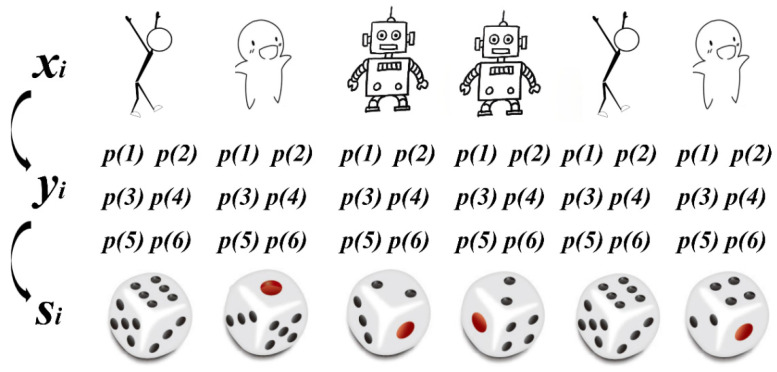
The decoding processing of Hidden Markov Model schematic.

**Figure 2 ijerph-19-14411-f002:**
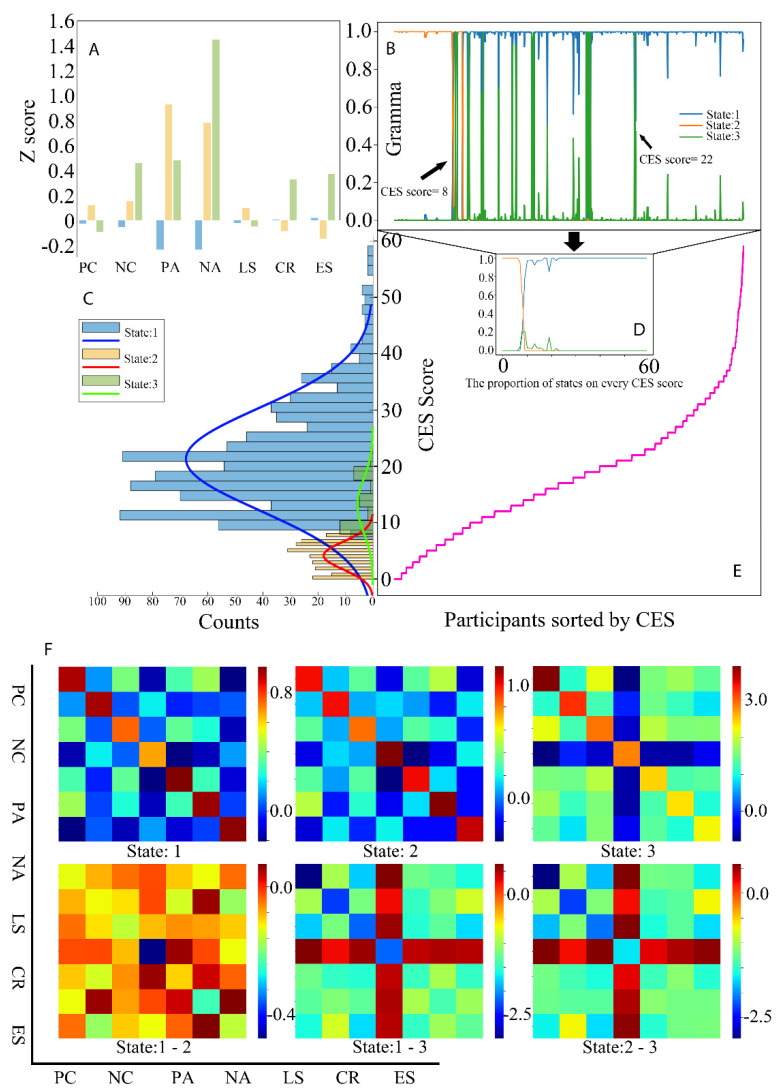
(**A**) shows the Z−scores of averaged observed variables among these three states. (**B**–**D**) showed the covered range without the intersection of states 1 and 2 are [0,7] and [23,60], respectively. State 2 has no exclusive covered range. (**E**) shows the CES score distribution of sorted participants. (**F**) shows the relationships of these observed variables in these three states by comparing among these three states. CES: the Chinese version Center for Epidemiological Studies Depression Scale; PC: positive coping; NC: negative coping; PA: positive affect; NA: negative affect; LS: life satisfaction; CR: cognitive reappraisal; ES: expression suppression.

**Figure 3 ijerph-19-14411-f003:**
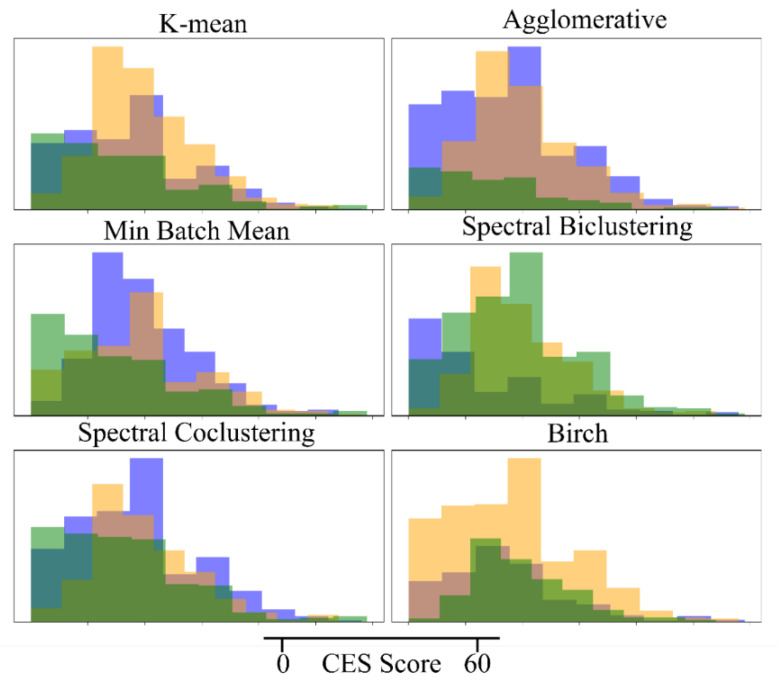
The effect of other kinds of clustering models in detecting the depression risk.

**Table 1 ijerph-19-14411-t001:** Socio-Demographic characteristics of the participants (*n* = 1128).

Characteristics	Items	*n*	%
Gender	Male	480	42.55
	Female	720	63.83
Age	Mean (std) = 19.86 (1.31)		
Number of siblings	0	670	59.40
	1	281	24.91
	2	177	15.69
Place of residence	City	490	43.44
	Town	638	56.56
Education of Father	Primary School	194	17.20
	Middle School	444	39.36
	High School	274	24.29
	Undergraduate	489	43.35
	Graduate	27	2.39
Education of Mother	Primary School	300	26.60
	Middle School	433	38.39
	High School	234	20.74
	Undergraduate	144	12.77
	Graduate	17	1.51

**Table 2 ijerph-19-14411-t002:** The Correlation among all variables.

All	Var	M (SD)	1	2	3	4	5	6	7	8
1	CES	18.000 (10.58)	—							
2	PC	2.943 (0.50)	−0.024	—						
3	NC	2.270 (0.51)	−0.050	0.142 ***	—					
4	PA	29.016 (7.60)	−0.223 ***	0.439 ***	0.115 ***	—				
5	NA	22.879 (7.72)	−0.240 ***	−0.172 ***	0.247 ***	0.234 ***	—			
6	LS	17.211 (5.74)	0.033	0.346 ***	0.030	0.397 ***	−0.231 ***	—		
7	CR	22.188 (3.74)	0.020	0.477 ***	0.016	0.269 ***	−0.175 ***	0.286 ***	—	
8	ES	11.326 (3.07)	0.009	−0.181 ***	0.087 **	−0.126 ***	0.111 ***	−0.095 **	0.010	—
Low-depression-risk group			1	2	3	4	5	6	7	8
1	CES	4.868 (2.78)	—							
2	PC	2.99 (0.50)	−0.050	—						
3	NC	2.339 (0.53)	−0.082	0.099	—					
4	PA	35.23 (7.53)	−0.112	0.396 ****	0.141 *	—				
5	NA	28.128 (8.76)	−0.171 **	−0.170 **	0.228 ***	0.207 **	—			
6	LS	17.811 (5.61)	0.010	0.342 ***	0.073	0.397 ***	−0.250 ***	—		
7	CR	22.037 (3.88)	0.128 *	0.450 ***	−0.105	0.248 ***	−0.178 **	0.179 **	—	
8	ES	10.992 (3.22)	−0.014	−0.178 **	0.189 **	−0.047	0.175 **	−0.118	−0.098	—
High-depression-risk group			1	2	3	4	5	6	7	8
1	CES	29.007 (7.62)	—							
2	PC	2.963 (0.49)	−0.096	—						
3	NC	2.267 (0.49)	−0.064	0.238 ***	—					
4	PA	28.017 (6.48)	0.030	0.495 ***	0.129 **	—				
5	NA	21.265 (6.15)	0.092	−0.171***	0.301 ***	0.102 *	—			
6	LS	17.698 (5.30)	0.004	0.316 ***	0.048	0.443 ***	−0.173 ***	—		
7	CR	22.19 (3.60)	−0.048	0.455 ***	0.028	0.320 ***	−0.191 ***	0.322 ***	—	
8	ES	11.35 (2.96)	−0.054	−0.221 ***	0.035	−0.209 ***	0.079	−0.047	0.051	—

Note: **** *p* < 0.0001, *** *p* < 0.001, ** *p* < 0.01, * *p* < 0.05. CES: the Chinese version Center for Epidemiological Studies Depression Scale; PC: positive coping; NC: negative coping; PA: positive affect; NA: negative affect; LS: life satisfaction; CR: cognitive reappraisal; ES: expression suppression.

**Table 3 ijerph-19-14411-t003:** Regression Analysis of Variables Relationship.

Var	β	t	95%CI		Multicollinearity	
			Lower	Upper	Tolerance	VIF
PC	0.01	0.19	−1.4	1.69	0.59	1.69
NC	0.01	0.43	−0.96	1.5	0.89	1.12
PA	−0.22	−6.02 ***	−0.41	−0.21	0.59	1.69
NA	−0.17	−4.95 ***	−0.32	−0.14	0.69	1.44
LS	0.07	2.16 *	0.01	0.26	0.72	1.40
CR	0.03	0.77	−0.11	0.26	0.73	1.37
ES	0.01	0.23	−0.18	0.22	0.93	1.08
All: F (7,1119) =16.282 ***, R^2^ = 0.087						
PC	−0.13	−1.59	−1.57	0.17	0.64	1.56
NC	−0.01	−0.15	−0.75	0.65	0.87	1.15
PA	−0.09	−1.07	−0.09	0.03	0.64	1.57
NA	−0.14	−1.84	−0.09	0.00	0.72	1.38
LS	0.02	0.28	−0.06	0.08	0.71	1.42
CR	0.18	2.44 *	0.02	0.23	0.76	1.32
ES	0.01	0.08	−0.11	0.12	0.91	1.1
low-depression-risk: F (7,235) =2.252, R^2^ = 0.035						
PC	−0.12	−1.77	−3.82	0.20	0.56	1.80
NC	−0.07	−1.33	−2.76	0.53	0.81	1.24
PA	0.06	0.89	−0.08	0.21	0.59	1.71
NA	0.10	1.76	−0.01	0.26	0.75	1.33
LS	0.03	0.57	−0.12	0.21	0.73	1.37
CR	0.00	0.02	−0.24	0.24	0.71	1.40
ES	−0.07	−1.37	−0.45	0.08	0.89	1.13
high-depression-risk: F (7,403) =1.825, R^2^ = 0.014						

Note: *** *p* < 0.001, * *p* < 0.05. PC: positive coping; NC: negative coping; PA: positive affect; NA: negative affect; LS: life satisfaction; CR: cognitive reappraisal; ES: expression suppression.

**Table 4 ijerph-19-14411-t004:** Independent samples *t*-test between risk group and nonrisk group.

Var	Group	M (SD)	*t* (df)	*p* Value	95%CI	
					Lower	Upper
CES	Risk	31.185 (7.31)	61.947 (411.88)	0.000	26.487	28.223
	NonRisk	3.83 (2.27)				
PC	Risk	2.952 (0.5)	−1.321 (505)	0.187	−0.149	0.029
	NonRisk	3.012 (0.48)				
NC	Risk	2.257 (0.49)	−1.913 (505)	0.056	−0.177	0.002
	NonRisk	2.344 (0.5)				
PA	Risk	28.047 (6.49)	−12.945 (505)	0.000	−8.970	−6.606
	NonRisk	35.835 (6.63)				
NA	Risk	21.285 (6.14)	−10.765 (310.65)	0.000	−8.798	−6.079
	NonRisk	28.723 (8.22)				
LS	Risk	17.777 (5.25)	0.087 (505)	0.931	−0.937	1.024
	NonRisk	17.734 (5.71)				
CR	Risk	22.219 (3.65)	1.113 (505)	0.266	−0.294	1.063
	NonRisk	21.835 (3.93)				
ES	Risk	11.245 (2.89)	1.309 (505)	0.191	−0.179	0.891
	NonRisk	10.888 (3.07)				

Note: CES: the Chinese version Center for Epidemiological Studies Depression Scale; PC: positive coping; NC: negative coping; PA: positive affect; NA: negative affect; LS: life satisfaction; CR: cognitive reappraisal; ES: expression suppression.

## Data Availability

Because personal privacy is involved and some of the subjects do not want their options to be actively disclosed. All data, scripts, and images in the present study are available to require.

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
