# Peer review of "A Depression-Risk Mental Pattern Identified by Hidden Markov Model in Undergraduates"

_ijerph, 2022, doi:10.3390/ijerph192114411_

Round 1
Reviewer 1 Report
Dear editors,
Thank you for asking me to review this paper. In general, the article is very original and interesting, especially for the purpose of your research. I really appreciate the preventive approach to depressive symptoms, but I think it needs a lot of editing before it is published.
In my opinion, the introduction lacks a detailed description of the panorama of the studies already present in the literature and is therefore too superficial. For example, what are the sources on which the explanations given in the first part of the introduction are based? I have seen that many sources are mentioned in the section of the discussions: normally in this section we discuss the results obtained by comparing them with the literature presented in the introduction, but in your case there is some confusion about this. Therefore I recommend to review both sections in a clear and orderly manner .
Furthermore, I suggest that more information should be given about the characteristics of the sample and that the tools used be described in more detail in order to clarify the methodology adopted.
Finally, I also recommend that you create a separate paragraph describing the limitations of your study.
I hope my suggestions can enrich and improve your interesting research.
Author Response
Dear Reviewer 1:
Thank you for your comments concerning our manuscript entitled “A Depression-Risk Mental Pattern Identified by Hidden Markov Model in Undergraduates” (ID: ijerph-1988865). Those comments are all valuable and very helpful for revising and improving our paper. We have studied comments carefully and have made the correction which we hope meet with approval. The revised portion is marked in red in the paper. Please see below for authors’ responses to your comments.
1.In my opinion, the introduction lacks a detailed description of the panorama of the studies already present in the literature and is therefore too superficial. For example, what are the sources on which the explanations given in the first part of the introduction are based? I have seen that many sources are mentioned in the section of the discussions: normally in this section we discuss the results obtained by comparing them with the literature presented in the introduction, but in your case there is some confusion about this. Therefore I recommend to review both sections in a clear and orderly manner.
- Response:Thank you very much for helpful comments, which are indeed very helpful. We have revised the introduction and discussion according to your suggestion and marked in red in the new manuscript.
- Furthermore, I suggest that more information should be given about the characteristics of the sample and that the tools used be described in more detail in order to clarify the methodology adopted.
- Response:Thanks for pointing this out. We have added a table per you suggestion. Please refer to the revisions in our new manuscript on table 1.
- Finally, I also recommend that you create a separate paragraph describing the limitations of your study.
- Response:Thanks for pointing this out. We have created a separate paragraph describing the limitations on page 12.

Reviewer 2 Report
Based on a sample of 1247 undergraduates, this paper aimed to identify the best easy access screening approach for indirectly predicting depression risks in this population.
1. Abstract section should be include more sample information (e.g., descriptive statistics). I encourage authors to review this aspect in the Abstract.
2. I recommend to add the term “subjective well-being” to keywords.
3. In the introduction section, it is missing that the authors justify the election of the study variables (motion regulation, subjective well-being and coping styles) and it is missing the literature review focused on these variables and their association with depression in undergraduate population.
4. I recommend authors include a section “the present study”. This section should be in another paragraph. Please, make sure all objectives are stated in this section and explain the results expected.
5. Please, describe the characteristics of the sample in more detail in terms of socioeconomic status, maybe create a new table would be useful.
6. Please, follow the same order in everyone measure and indicate in clear way the subscales number and an example item for every subscale. Moreover, for all measures used, please indicate whether response scores were summed or averaged to create their composite scores for data analysis.
7. Discussion section should be start with the main purpose of the study. Overall, I recommend authors to review and reorganized the Discussion and conclusion sections. The discussion is underdeveloped. It would be important for the authors to work on connecting the information on the Discussion with the Introduction, integrating the interpretation of the findings. So, I recommend authors review this entire section in base on changes I propose in Introduction section. This section should be end with the limitations and strengths of the study. Besides, the Conclusion section should be included the main conclusions of the study and practical implications or future work.
8. Please, revise according to 7th Edition APA style the entire manuscript (e.g., Recommended Verb Tenses in APA Style Papers, Recommended Verb Tenses in APA Style Papers, Style References…).
Author Response
Dear Reviewer 2:
Thank you for your comments concerning our manuscript entitled “A Depression-Risk Mental Pattern Identified by Hidden Markov Model in Undergraduates” (ID: ijerph-1988865). Those comments are all valuable and very helpful for revising and improving our paper. We have studied comments carefully and have made the correction which we hope meet with approval. The revised portion is marked in red in the paper.
- Abstract section should be include more sample information (e.g., descriptive statistics). I encourage authors to review this aspect in the Abstract.
- Response:Thanks for pointing this out. We have added the descriptive statistics of sample in the Abstract and marked in red.
- I recommend to add the term “subjective well-being” to keywords.
- Response:Thank you for your helpful comments. We have added the term “subjective well-being” to keywords.
- In the introduction section, it is missing that the authors justify the election of the study variables (motion regulation, subjective well-being and coping styles) and it is missing the literature review focused on these variables and their association with depression in undergraduate population.
- Response:Thank you very much for helpful comments, which are indeed very helpful to improve our manuscript. We have revised the introduction according to your suggestion and marked in red in the manuscript.
- I recommend authors include a section “the present study”. This section should be in another paragraph. Please, make sure all objectives are stated in this section and explain the results expected.
- Response:Thanks for pointing this out. We have added another paragraph entitled “the present study” in the introduction per you suggestion. In this paragraph, goals and expectations were stated. Please refer to the revisions in on page 3.
- Please, describe the characteristics of the sample in more detail in terms of socioeconomic status, maybe create a new table would be useful.
- Response:Thank you very much for helpful comments. We have added a table per you suggestion. Please refer to the revisions in our new manuscript on table 1.
- Please, follow the same order in everyone measure and indicate in clear way the subscales number and an example item for every subscale. Moreover, for all measures used, please indicate whether response scores were summed or averaged to create their composite scores for data analysis.
- Response:Thanks for pointing this out. We have added the information for every subscale per you suggestion.
- Discussion section should be start with the main purpose of the study. Overall, I recommend authors to review and reorganized the Discussion and conclusion sections. The discussion is underdeveloped. It would be important for the authors to work on connecting the information on the Discussion with the Introduction, integrating the interpretation of the findings. So, I recommend authors review this entire section in base on changes I propose in Introduction section. This section should be end with the limitations and strengths of the study. Besides, the Conclusion section should be included the main conclusions of the study and practical implications or future work.
- Response:Thank you very much for helpful comments, which are indeed very helpful. We have revised the discussion according to your suggestion and marked in red in the new manuscript. Also, we added the practical implications and future work in conclusion.
- Please, revise according to 7th Edition APA style the entire manuscript (e.g., Recommended Verb Tenses in APA Style Papers, Recommended Verb Tenses in APA Style Papers, Style References…).
- Response:Thanks for pointing this out. We have checked the verb tenses of the whole article, and revised the verb tenses in APA Style Papers. The revised verbs were marked in red in the new manuscript.

Round 2
Reviewer 1 Report
Dear authors,
I am pleased that you have made the changes I suggested. I find that the new form of the article, more complete and rich in details, is much more interesting and of particular importance for the scientific panorama.
Thanks you very much.